# Safe Learning of PDDL Domains with Conditional Effects

**Primary Keywords:**

*(2) Learning; (8) Knowledge Representation/Engineering*

## Abstract

Powerful domain-independent planners have been developed to solve various types of planning problems. These planners often require a model of the acting agent's actions, given in some planning domain description language. Manually designing such an action model is a notoriously challenging task. An alternative is to automatically learn action models from observation. Such an action model is called safe if plans consistent with it are also consistent with the real, unknown action model. Algorithms for learning such safe action models exist, yet they cannot handle domains with conditional or universal effects, which are common constructs in many planning problems. We prove that learning non-trivial safe action models with conditional effects may require an exponential number of samples. Then, we identify reasonable assumptions under which such learning is tractable and propose SAM Learning of Conditional Effects (Conditional-SAM) the first algorithm capable of doing so. We analyze this Conditional-SAM theoretically and evaluate it experimentally. Our results show that the action models learned by Conditional-SAM can be used to solve perfectly most of the test set problems in most of the experimented domains.

## 1 Introduction

Planning is the fundamental task of choosing which actions to perform in order to achieve a desired outcome. An *automated domain-independent planner* refers to an Artificial Intelligence (AI) algorithm capable of solving a wide range of planning problems (Ghallab, Nau, and Traverso 2016). Developing a domain-independent planner is a long-term goal of AI research. Researchers developed many domain-independent planners for various types of planning problems. Such planners include Fast Downward (Helmert 2006), Fast Forward (Hoffmann 2001), ENHSP (Scala et al. 2016), and more. These planners require a model of the acting agent's actions, given in some domain description language such as the Planning Domain Definition Language (**PDDL**) (Ghallab et al. 1998)). Defining an agent's *action model* to solve real-world problems is extremely hard. Researchers acknowledged this modeling challenge and algorithms for learning action models from observations have been proposed (Cresswell and Gregory 2011; Aineto, Celorrio, and Onaindia 2019; Yang, Wu, and Jiang 2007; Juba, Le, and Stern 2021; Mordoch, Stern, and Juba 2023). Since the learned model may differ from the domain's actual action model, using it to

plan raises two challenges: *completeness* and *soundness*. The *completeness* challenge is that the learned model may be too *restrictive*, in the sense that it does not allow generating plans for solvable planning problems. The *soundness* challenge is that the learned model may be too *permissive*, in the sense that it allows plans that either cannot be applied in the domain or do not reach a state that satisfies the problem goals. There is an inherent trade-off between the soundness and completeness of the learned model. In problem settings where execution failures are unacceptable or are very costly, e.g., autonomous vehicles, high-end robotics, and medical treatment planning, soundness becomes a hard constraint. We focus on such cases, and aim to learn an action model that satisfies the strongest form of soundness: every plan allowed by the learned model must be applicable and yield the same states in the real, unknown, model. An action model that satisfies this requirement has been called *safe* (Juba, Le, and Stern 2021; Juba and Stern 2022; Mordoch, Stern, and Juba 2023).[1] We view this as a "safety" notion in part since it enables more conventional notions of safety to be enforced during planning, and provides assurance that they will carry over to the actual execution.

Algorithms from the Safe Action Model Learning (SAM) family (Stern and Juba 2017; Juba, Le, and Stern 2021; Juba and Stern 2022; Mordoch, Stern, and Juba 2023) address the challenge of learning safe action models under different sets of assumptions. However, these algorithms are not suitable for learning actions that may include *conditional effects*. A conditional effect is an effect that occurs only when a specific condition holds. For example, consider an AI for planning treatments to patients and the action of giving a flu medicine to a patient, where that medicine causes an allergic reaction in patients with a certain rare blood type. This action's effects include not having the flu, but there is also a conditional effect specifying that an allergic reaction occurs if the patient has a rare blood type. Clearly, if the patient has a rare blood type, we would want to avoid applying this type of treatment. A safe action model would never permit the execution of the action in these cases.

Previous works on action model learning with conditional

---

[1]This notion of safety has different definitions in different contexts. For example, safe Reinforcement Learning often refers to ensuring that some safety function of the current state never goes below some threshold value (Wachi and Sui 2020).

effects (Oates and Cohen 1996; Zhuo et al. 2010) made no safety guarantees for the learned models. This work addresses this gap by exploring the problem of learning safe action models for PDDL (Ghallab et al. 1998) domains with conditional effects. Specifically, we introduce the Conditional-SAM algorithm, which is guaranteed to outputs a safe action model. We show that Conditional-SAM requires an asymptotically optimal number of trajectories when the size of the antecedents for the conditional effects is restricted, which is the only case where the problem is tractable. Then, we describe how Conditional-SAM can be extended to support *lifted action models* (i.e., parameterized) and effects with *universally quantified variables*. Finally, we demonstrate the usefulness of Conditional-SAM in practice on a set of benchmark planning problems that have conditional and universal effects. Our results show that given a few observations, the model Conditional-SAM learns is logically identical to the real action models and can be used to solve test problems for most of the experimented domains.

## 2 Preliminaries

We focus on planning problems in domains where action outcomes are deterministic, the states are fully observable, and contain Boolean variables only. Such problems are commonly modeled using a fragment of the ADL (Action Description Language) (Pednault 1989) and formulated in PDDL (Planning Domain Definition Language) (Ghallab et al. 1998). In PDDL, a planning problem is described by a PDDL domain and a PDDL problem. A PDDL domain is a tuple $D = \langle F, A \rangle$ where $F$ is a finite set of Boolean variables, referred to as fluents and $A$ is a set of actions. A *literal* refers to either a fluent or its negation. Let $L$ be the set of every possible literal. A *state* is a conjunction of literals that includes, for every fluent $f$, either $f$ or $\neg f$. The value of a fluent $f$ is a state $s$ is true if $s$ includes $f$ and false otherwise. An action $a \in A$ is a triple $\langle name(a), pre(a), eff(a) \rangle$ corresponding to the name of the action, its preconditions, and effects. The preconditions of an action $a$, $pre(a)$, is a conjunction of literals that are sufficient and necessary conditions for applying $a$. If the preconditions of $a$ are satisfied in a state $s$ we say that $a$ is applicable in $s$. If an action has no preconditions, then it is applicable in any state. The effects of an action $a$, $eff(a)$, specify the outcome of applying $a$. An effect is defined by a tuple $\langle c, e \rangle$ where $c$ is called the antecedent (condition) and $e$ is called the result. Both $c$ and $e$ are conjunctions of literals. The semantics of an effect $\langle c, e \rangle$ for an action $a$ is that if $a$ is applied in a state $s$ where the antecedent $c$ holds then the result $e$ will be true in the next state. The antecedent can also be *true*, representing that the result occurs regardless of the state where the action has been applied. An effect where the antecedent is not *true* is called a *conditional effect*. The outcome of applying $a$ to a state $s$, denoted $a(s)$, is a state in which the value of every fluent is as in state $s$ except those fluents changed by the action's effects.

A PDDL *problem* is defined by a tuple $P = \langle I, G, D \rangle$ where $I$ is the initial state of the world, $G$ is a conjunction of literals that define the desired goal, and $D$ is a PDDL domain. A *plan* $\Pi = \langle a_1, a_2, ...a_n \rangle$ is a sequence of actions. A plan $\Pi$ is called *valid* for a PDDL problem $P = \langle I, G, D \rangle$ if $a_1$ is applicable in $I$, $a_i$ is applicable in $a_{i-1}(\cdots(a_1(I))\cdots)$, and $G \subseteq a_n(a_{n-1}(\cdots(a_1(I))\cdots))$.

An *action model* for a PDDL domain $D = \langle F, A \rangle$ is a pair $M = \langle pre_M, eff_M \rangle$ where $pre_M$ maps every action in $A$ to a (possibly empty) conjunction of literals in $F$, and $eff_M$ maps actions in $A$ to a (possibly empty) set of effects over $F$. The *real action model* of a domain, denoted $M^*$, is the action model where for every action $a$ is mapped to its real precondition and effects, i.e., $pre_{M^*}(a) = pre(a)$ and $eff_{M^*}(a) = eff(a)$. For an action $a$, state $s$, and action model $M$, we denote by $a_M(s)$ the state that results from applying $a$ in $s$ assuming that $M$ is the real action model.

**Definition 1** (Safe Action Model). *An action model $M$ is safe w.r.t. an action model $M'$ if for every state $s$ and action $a$ it holds that if $a$ is applicable in $s$ according to $M$ then (1) it is also applicable in $s$ according to $M'$, and (2) applying $a$ in $s$ results in exactly the same state according to both $M$ and $M'$. Formally:*

$$pre_M(a) \subseteq s \rightarrow \big(pre'_M(a) \subseteq s \wedge a_M(s) = a'_M(s)\big) \quad (1)$$

An action model is said to be safe in a domain if it is safe w.r.t. its real action model. This paper deals with the case where the planning agent does not know the real action model of a given domain, yet it aims to learn an action model that is safe in it. A major benefit of learning such a safe action model is that any plan generated with the learned action model for any problem in the same domain is also valid with respect to the real, unknown, action model. Following prior works on learning action models (Amir and Chang 2008; Cresswell, McCluskey, and West 2013; Aineto, Jiménez, and Onaindia 2018) in general and safe action models in particular (Stern and Juba 2017; Juba, Le, and Stern 2021; Mordoch, Stern, and Juba 2023), we assume as input a set of observations of previously executed plans, represented as a set of *trajectories*. A trajectory $T = \langle s_0, a_1, s_1, \ldots a_n, s_n \rangle$ is an alternating sequence of states $(s_0, \ldots, s_n)$ and actions $(a_1, \ldots, a_n)$ that starts and ends with a state. The trajectory created by applying $\pi$ to a state $s$ is the sequence $\langle s_0, a_1, \ldots, a_{|\pi|}, s_{|\pi|} \rangle$ such that $s_0 = s$ and for all $0 < i \leq |\pi|$, $s_i = a_i(s_{i-1})$. A trajectory is often represented as a set of *action triplets* $\big\{ \langle s_{i-1}, a_i, s_i \rangle \big\}_{i=1}^{|\pi|}$.

### Problem Definition and Assumptions

We deal with the problem of learning a safe action model for a domain $D$ given a set of trajectories $\mathcal{T}$ collected by executing plans for different problems in $D$. Ideally, the learned action model will be able to generalize beyond the given set of trajectories and enable finding plans for other problems in $D$. We make the following assumptions:

1. The given trajectories are fully observable and noise-free.
2. For each literal $l'$ and action $a$, there is *at most* one effect of $a$ for which $l'$ is a result.
3. The maximal number of literals in an antecedent is at most $n$, a fixed parameter known in advance.

Assumption 1 means we observe all the actions and the values of all the fluents in all states in every trajectory in $\mathcal{O}$. This assumption is common in the action model learning literature and lifting it in the context of conditional effects

is left for future work. Assumption 2 means that there are no *disjunctive antecedents*, i.e., multiple effects for the same action having the same result but with different antecedents. The implication of this assumption is that if $(c, e)$ is an effect of some action $a$, then no conjunction of literals except $c$ is an antecedent of the literals in the result $e$. Formally:

$$\forall l', a : ((c, e) \in \mathit{eff}(a) : l' \in e)$$
$$\rightarrow (\nexists (c', e') \in \mathit{eff}(a) : l' \in e' \wedge c \neq c') \quad (2)$$

This assumption crucially improves the efficiency of learning. We discuss relaxing this assumption later. Assumption 3 means that a human modeler must specify an upper bound on the number of literals in an antecedent for the domain at hand. Specifying such a bound is significantly easier than manually defining the entire action model. We prove later that without the third assumption, learning conditional effects is intractable.

## 3 Approach

Conditional-SAM learns an action model by applying the following rules:

**Definition 2** (Conditional-SAM Inductive Rules). *For every action triplet $\langle s, a, s' \rangle \in \mathcal{T}$*
1. *[Not a precondition] For every literal $l \notin s$, $l \notin \mathit{pre}(a)$*
2. *[Not a result] For every literal $l' \notin s'$, $\nexists (c, e) \in \mathit{eff}(a)$ where $(c \wedge s \nvdash \bot) \wedge (l' \in e)$*
3. *[Must be an effect] For every literal $l' \in s' \setminus s$, $\exists (c, e) \in \mathit{eff}(a) : (c \wedge s \nvdash \bot) \wedge (l' \in e)$*
4. *[Not an antecedent] For every literal $l' \in s' \setminus s$, and conjunction of literals $c$: if $c \wedge s \vdash \bot$, then $\nexists (c', e) \in \mathit{eff}(a)$ such that $c \subseteq c'$ and $l' \in e$.*

The first three rules generalize the SAM-Learning (Juba, Le, and Stern 2021) inductive rules to support conditional effects. The fourth inductive rule is derived from Assumption 2 (no disjunctive conditional effects): if we observe $l'$ as the result of the action, any conjunction of literals $c$ that is not satisfied in $s$ cannot be the antecedent of the conditional effect for $l'$.

**Example 1.** *Consider a domain with three fluents $f_1$, $f_2$, and $f_3$, where the size of antecedents is bounded by 1 (i.e., $n = 1$), and assume a Boolean vector of size 3 represents a state. Now, assume we observed an action triplet $\langle (T, T, F), a, (F, T, F) \rangle$. Using the first inductive rule, we infer that $\neg f_1$, $\neg f_2$, and $f_3$ are not preconditions of the action $a$. By applying the second inductive rule, $a$ cannot include an effect $(c, e)$ such that $c$ is consistent with $f_1 \wedge f_2 \wedge \neg f_3$ and the result is either $f_1$, $\neg f_2$, or $f_3$. Since $n = 1$, this rules out the conditional effects where $c$ is one of the following $\{true, f_1, f_2, \neg f_3\}$ and $e$ is either $f_1$ or $\neg f_2$ or $f_3$, e.g., $(c, e) = (f_1, \neg f_2)$. According to the third inductive rule there exists $(c, e) \in \mathit{eff}(a)$ such that $c$ is one of $\{true, f_1, f_2, \neg f_3\}$ and $e = \neg f_1$. Finally, according to the fourth inductive rule $(\neg f_2, \neg f_1)$ and $(f_3, \neg f_1)$ cannot be conditional effects.*

### Conditional-SAM Algorithm

Next, we describe the Conditional-SAM algorithm, which uses the Conditional-SAM inductive learning rules (Definition 2). The pseudo-code for Conditional-SAM is given in

Algorithm 1. Let $A(\mathcal{T}), L(\mathcal{T})$ be the set of actions and literals observed in the trajectories $\mathcal{T}$. Conditional-SAM maintains three data structures: $\mathit{pre}(a)$ and $\mathit{PosAnte}(\cdot, a)$ for every action $a$, and $\mathit{MustBeResult}(a, l)$ for action $a$ and literal $l$. $\mathit{pre}(a)$ is a set of literals, representing which literals may be preconditions of $a$. It is initialized to all the literals $l \in L(\mathcal{T})$. $\mathit{PosAnte}(l, a)$ is a set of conjunctions of literals, representing all the conjunctions that may be antecedents of a conditional effect of $a$ that results in $l$.[2] This data structure is initialized to include every conjunction of literals of size $n$ or less. *MustBeResult*$(a)$ maintains the set of literals observed to be a result of applying $a$. This data structure is initialized as an empty set. Conditional-SAM updates these data structures by applying the Conditional-SAM inductive learning rules for each action triplet in the given trajectories. That is, it removes literals from $\mathit{pre}(\cdot)$ according to Rule 1, removes conjunctions of literals from $\mathit{PosAnte}(l, a)$ using Rules 2 and 4, and adds literals to $\mathit{MustBeResult}(a)$ using Rule 3.

Then, Conditional-SAM iterates over every action $a \in A(\mathcal{T})$ using $\mathit{pre}(a)$, $\mathit{PosAnte}(\cdot, a)$, and $\mathit{MustBeResult}(a)$ to generate the preconditions and effects of $a$ in the resulting safe action model. This part of Conditional-SAM is encapsulated in the function *BuildActionModel*, listed in Algorithm 2. BuildActionModel stores the preconditions and effects of the resulting safe action model in $\mathit{pre}^*(a)$ and $\mathit{eff}^*(a)$, respectively. Initially, $\mathit{eff}^*(a)$ is an empty set and $\mathit{pre}^*(a)$ is set to be $\mathit{pre}(a)$. Then, it iterates over every literal $l$ and considers adding an effect to $\mathit{eff}^*(a)$ with a $l$ as a result, as follows. Let $PA$ be the subset of $\mathit{PosAnte}(l, a)$ containing only conjunctions that are disjoint from $\mathit{pre}(a)$. Conditional-SAM uses $PA$ to compute two formulas, *Ante* and *NotAnte*. *NotAnte* is the conjunction of the negation of every clause $c$ in $PA$, and *Ante* is the conjunction of all the clauses $c \in PA$. Observe that applying $a$ in a state where *Ante* is true guarantees that $l$ will be true in the subsequent state. Similarly, applying $a$ in a state where *NotAnte* is true guarantees that $l$ will not be true in the subsequent state unless it was true before. To minimize the number of clauses and their size, we apply unit propagation on each of them. Afterward, the function verifies whether $l \in \mathit{MustBeResult}(a)$. If so, the tuple $(\mathit{Ante}, l)$ is added to $\mathit{eff}(a)$. If $\mathit{PosAnte}(l, a)$ includes more than a single clause of possible antecedents, then there is an ambiguity on which antecedent causes $l$. To mitigate this, Conditional-SAM adds to $\mathit{pre}^*(a)$ the disjunction $(l \vee \mathit{NotAnte} \vee \mathit{Ante})$. This disjunction is composed of three parts as follows: First, allowing the action to be applicable if the result, $l$, is observed in the pre-state. Second, the action is permitted if *none* of the antecedents hold in the pre-state. Last, $a$ is applicable if *all* the antecedents hold in the pre-state. If one of the above holds, the action can be executed.

If $l$ was not observed as a result of the action, i.e., $l \notin \mathit{MustBeResult}(a)$, the function adds $(l \vee \mathit{NotAnte})$ to $\mathit{pre}^*(a)$. Since we have yet to observe $l$ as a result of the action, then $l \notin \mathit{eff}(a)$; Thus, to prevent $l$ from triggering unexpectedly,

_______________

[2] According to Assumption 2, in the real action model there can be only one such conjunction. Conditional-SAM maintains in $\mathit{PosAnte}(l, a)$ a set of conjunctions since it does not know the real action model.

we do not permit the action to be executed if *Ante* is true. After repeating this for every action, *BuildActionModel* returns the safe action model comprising $pre^*$ and $eff^*$.

---

**Algorithm 1: Conditional-SAM Algorithm**

---
1: **Input**: $\mathcal{T}, n$
2: **Output**: A safe action model.
3: **for** $a \in A(\mathcal{T})$ **do**
4:     $pre(a) \leftarrow L(\mathcal{T})$
5:     $MustBeResult(a) \leftarrow \emptyset$
6:     $PosAnte(l, a) \leftarrow \bigcup_{i=1}^{n} \{l_1 \wedge ... \wedge l_i | \forall 1 \leq j \leq i : l_j \in L(\mathcal{T})\} \cup \{true\}$
7: **for** $\langle s, a, s' \rangle \in \mathcal{T}$ **do**
8:     **for** $l$ such that $l \notin s$ **do**
9:         $pre(a) \leftarrow pre(a) \setminus \{l\}$                    ▷ Rule 1
10:     **for** $l \in s' \setminus s$ **do**                          ▷ Rule 3
11:         $MustBeResult(a) \leftarrow MustBeResult(a) \cup \{l\}$
12:     **for** $l' \notin s'$ and $c \in PosAnte(l', a)$ s.t. $(c \wedge s \nvdash \bot)$ **do**
13:         $PosAnte(l', a) \leftarrow PosAnte(l', a) \setminus c$
                                                            ▷ Rule 2
14:     **for** $l' \in s' \setminus s$ and $c \in PosAnte(l', a)$ s.t. $c \wedge s \vdash \bot$ **do**   ▷ Rule 4
15:         $PosAnte(l', a) \leftarrow PosAnte(l', a) \setminus c$
16: **return BuildActionModel**$(pre, MustBeResult, PosAnte)$

---

---

**Algorithm 2: BuildActionModel**

---
1: **Input**: $pre, MustBeResult, PosAnte$
2: **Output**: $pre^*$ and $eff^*$ for all actions.
3: **for** $a \in A(\mathcal{T})$ **do**
4:     $eff(a) \leftarrow \emptyset$;   $pre^*(a) \leftarrow \bigwedge_{l \in pre(a)} l$
5:     **for** $l \in L(\mathcal{T}) \setminus pre(a)$ where $PosAnte(l, a) \neq \emptyset$ **do**
6:         $PA \leftarrow \{c \in PosAnte(l, a) | (pre(a) \cap c) = \emptyset\}$
7:         $NotAnte \leftarrow \bigwedge_{c \in PA} \neg c$
8:         $Ante \leftarrow \bigwedge_{c \in PA} c$
9:         Minimize clauses $Ante$ and $NotAnte$ using unit propagation.
10:         **if** $l \in MustBeResult(a)$ **then**
11:             Add to $eff(a)$: $(Ante, l)$
12:             **if** $PA$ is not a single clause **then**
13:                 $pre^*(a) \leftarrow pre^*(a) \wedge (l \vee NotAnte \vee Ante)$
14:         **else**
15:             $pre^*(a) \leftarrow pre^*(a) \wedge (l \vee NotAnte)$
16: **return** $\langle pre^*, eff \rangle$

---

**Example 2.** *Given a domain with 3 literals and an action $a$ where $pre(a) = \emptyset$, $l_1 \in MustBeResult(a)$, $PosAnte(l_1, a) = \{\{l_2\}, \{l_3\}\}$, $PosAnte(l_2, a) = \emptyset$, and $PosAnte(l_3, a) = \emptyset$. The resulting preconditions and effects after applying BuildActionModel are $pre^*(a) = (l_1) \vee (\neg l_2 \wedge \neg l_3) \vee (l_2 \wedge l_3)$ and $eff^*(a) = (l_2 \wedge l_3, l_1)$, i.e., when $l_2 \wedge l_3$ then $l_1$.*

**Theorem 3.1.** *The action model $M'$ learned by Conditional-SAM is safe w.r.t the action model that generated the input trajectories $\mathcal{T}$.*

A proof is provided in the supplementary material. Note that the real model does not have actions with disjunctive antecedents (Assumption 2), but the safe action model we learn may include such actions. This highlights that the learned

model may be different from the real model. Nevertheless, the learned model is guaranteed to be safe w.r.t to it.

## 4 Theoretical Analysis

Next, we analyze the Conditional-SAM algorithm. We prove that under a fixed antecedent size ($n$) its space, runtime, and sample complexity are tractable, and show that our sample complexity bound is tight.

**Lemma 4.1.** *The space complexity of Conditional-SAM is $O\left(|A||L|^{n+1} \left(\frac{e}{n}\right)^n\right)$, where $e$ is the base of the natural logarithm.*

**Lemma 4.2.** *The runtime complexity of Conditional-SAM is $O\left(|A||L|^n \left(\frac{e}{n}\right)^n\right) + |\mathcal{T}||L|^{n+1} \left(\frac{e}{n}\right)^n\right)$*

As can be seen from Lemmas 4.1 and 4.2, the complexity of the algorithm is independent of the number of effects and is only affected by the maximal size of the antecedents and the number of literals and actions in the domain. The complexity does, however, increase exponentially with $n$.

**Theorem 4.3.** *Let $\mathcal{D}$ be a distribution over pairs $\langle P, \Pi \rangle$ where $P$ is a problem from a fixed domain $D$ and $\Pi$ is a plan solving $P$. Given*

$$m \geq \frac{1}{\epsilon}\left(\ln(3)|F||A| + 2\ln(2)|F||A|\left(\frac{2|F|e}{n}\right)^n + \ln\frac{1}{\delta}\right)$$

*trajectories obtained by executing $\Pi$ for $m$ independent draws from $\mathcal{D}$, Conditional-SAM returns an action model $M'$ such that with probability $1 - \delta$, for a new $P$ drawn from $\mathcal{D}$, the probability that there exists a plan consistent with $M'$ solving $P$ is at least $1 - \epsilon$.*

We supply the proofs for the space, runtime, and sample complexity in the supplementary material. At a high level, the sample complexity follows since the learned preconditions of each action of a plan $\Pi$ sampled from $\mathcal{D}$ are satisfied and the action model is safe. Either at least one literal is deleted from *pre* or at least one clause $c$ is deleted from some $PosAnte(e, a)$ when such an action $a$ would be used by the plan $\Pi$. Thus if a specific literal or clause would prohibit the action with probability greater than $\epsilon$, that literal/clause is eliminated with high probability given a sample of the specified size.

Conditional-SAM, therefore enjoys approximate completeness with high probability so long as the number of training trajectories is sufficiently large. The one unsatisfying aspect of our bound is that the number of trajectories is exponential in the size of the antecedents of the conditions in the conditional effects we consider. Unfortunately, we find that this is unavoidable and our bound is asymptotically optimal (for any fixed $n$) for safe action model learning for domains with conditional effects:

**Theorem 4.4.** *Any learning algorithm that is guaranteed to return a safe action model must be given at least $m \geq \Omega(\frac{1}{\epsilon}(|F||A||(|F|/3n)^n + \log\frac{1}{\delta}))$ samples to be able to guarantee that with probability at least $1 - \delta$ the learned model permits a plan solving $\Pi$ drawn from $\mathcal{D}$ with probability at least $1 - \epsilon$ for $0 < \epsilon, \delta < 1/4$.*

The full proof of the lower bound is in the supplemental material. At a high level, the hard distribution involves initial

states that have all $|A|$ "goal" fluents set to false, all but one (uniformly random) of the $(p - |A|)/2$ "forbidden" fluents true, and exactly $n$ out of the $(p - |A|)/2$ "flag" fluents (uniformly at random) true. With probability $4\epsilon$, the goal includes a single goal fluent, chosen uniformly at random, that should be set to true. All other goal fluents, as well as the one forbidden fluent, must be set to false. The corresponding $\Pi$ consists of a plan with a single action, where the agent takes an action corresponding to the fluent to be set true in the goal. Otherwise there is an empty goal, where the agent takes a no-op action. For any problem with a non-empty goal that we did not observe in the training set, no safe action model can permit taking the action needed to achieve the goal. We need to observe at least a $3/4$ fraction of the possible goals for a safe action model to attain probability $1 - \epsilon$.

## 5 Learning Lifted Action Models

It is common to define PDDL domains and problems in a *lifted* manner. A lifted domain defines fluents and actions in a parameterized manner, where every parameter has a *type*. For example, the action *(stop ?f - floor)* and the fluent *(destin ?person - passenger ?floor - floor)* from the IPC Miconic domain are parameterized by objects of type *floor* and *person*. A state is a conjunction of *grounded fluents*, which are pairs of the form $\langle l, b_l \rangle$ where $l$ is a fluent, and $b_l$ is a function that maps parameters of $l$ to concrete objects. A plan is a sequence of *grounded actions*, which are pairs in the form $\langle a, b_a \rangle$ where $a$ is an action and $b_a$ maps action parameters to objects. A trajectory is an alternating sequence of states and grounded actions.

Generally, the parameters in an action's preconditions and effects are bound to the action's parameters. Thus, preconditions and effects of an action in a lifted domain are *parameter-bound literals*. A parameter-bound literal for an action $a$ is a pair $(l, b_{la})$ where $l$ is a literal and $b_{la}$ is a function that maps every parameter of $l$ to a parameter in $a$. Let *bindings*$(a)$ be the function that returns all parameter-bound literals that can be bound to $a$. For a grounded action $a_G = \langle a, b_a \rangle$ and parameter-bound literal $l \in bindings(a)$, we define $g(a_G, l)$ to be the grounded literal resulting from assigning the objects in the parameters of $a_G$ to the parameters of $l$. Given a conjunction of parameter-bound literals $c$, $g(a_G, c)$ returns the corresponding conjunction of grounded literals $c_G$ such that $\forall l \in c : g(a_G, l) \in c_G$. Similarly, for a pair of conjunctions of parameter-bound literals $(c, e)$ we define $g(a_G, c, e)$ to be the pair $(c_G, e_G)$ that are the corresponding conjunctions of grounded literals. SAM learning has already been extended to learn lifted classical planning domains (Juba, Le, and Stern 2021) without conditional effects. We extend Conditional-SAM to support lifted domains in a similar manner, based on the following extension to the Conditional-SAM inductive rules (Def. 2).

**Definition 3** (Lifted Conditional-SAM Inductive Rules). *For every action triplet $\langle s, a_G = \langle a, b_a \rangle, s' \rangle \in \mathcal{T}$,*
1. *[Not a precondition] For every $l \in bindings(a)$ s.t. $g(a_G, l) \notin s$, $l \notin pre(a)$*
2. *[Not a result] For every $l' \in bindings(a)$ s.t. $g(a_G, l') \notin s'$, $\nexists(c, e) \in eff(a)$ where $(g(a_G, c) \wedge s \nvdash \bot) \wedge (l' \in e)$*

3. *[Must be an effect] For every $l' \in bindings(a)$, if $g(a_G, l') \in s' \setminus s$ then $\exists(c, e) \in eff(a)$, where $(g(a_G, c) \wedge s \nvdash \bot) \wedge (l' \in e)$*
4. *[Not an antecedent] For every $l' \in bindings(a)$ and set of literals $c \subseteq bindings(a)$ if $g(a_G, l') \in s' \setminus s$ and $g(a_G, c) \wedge s \vdash \bot$ then $\nexists(c', e) \in eff(a)$ such that $c \subseteq c'$ and $l' \in e$*

The rest of the Conditional-SAM algorithm remains essentially the same, where *MustBeResult* and *PosAnte* may now contain parameter-bound literals.

## 6 Learning Effects with Universal Quantifiers

Some PDDL domains and planners support *universal quantifiers* which allow actions' preconditions and effects to include additional parameters that are not bound to the actions' parameters. More formally, *Universally quantified* preconditions and effects define one or more *universally quantified variables (UQV)* that may be bound to any parameter of a literal used in them. The result of universally quantified conditional effects *must* include at least one UQV. Otherwise, if only the antecedents include UQVs, then we can interpret such antecedents as disjunctive universal preconditions. To clarify the formulation of universal effects, suppose that we need to represent an elevator with a stopping functionality that ensures that all waiting passengers get in or out once the elevator stops. Figure 1 presents the *stop* action schema to implement this functionality. The UQV in this example is ?p.

```
(:action stop
:parameters (?f - floor)
:precondition (and (lift-at ?f))
:effect
(and (forall (?p - passenger)
    (when
       (and (boarded ?p) (destin ?p ?f))
       (and (not (boarded ?p)) (served ?p))))))
```

Figure 1: Parts of the action *stop* from Miconic domain that contains universally conditional effects.

We focus below on learning universal effects since they are more common, but our approach also supports learning universal preconditions. Note that universal effects may be unconditional and might occur every time the action is executed, in which case the antecedent is the trivial antecedent *true*. Conditional-SAM can learn these universal effects as well. We briefly describe how Conditional-SAM can be extended to support universal effects.

In general, the number of UQVs a universal effect can define is exponential in the arity of the domain fluents. Still, universal effects with more than two UQV are rare. Thus, we will assume the number of UQVs in a universal effect is a known fixed constant $k$. To support universal effects, the *bindings*$(a)$ function is modified to also return parameter-bound literals that bind one or more literal parameters to UQVs that may be used in action $a$'s effects. Similarly, the $g(a_G, l)$ function is modified such that if $l$ is a parameter-bound literal that includes UQVs then $g(a_G, l)$ returns a *set* of grounded literals matching the grounded action's parameters

combined with the UQVs. In addition, $g(a_G, c, e)$ returns a set of matching pairs $(c_G, e_G)$ if either $c$ or $e$ include one or more UQVs. We now present the changes in the inductive rules to support universally quantified variables.

**Definition 4** (Conditional-SAM Inductive Rules with UQVs).
*For every action triplet $\langle s, a_G = \langle a, b_a \rangle, s' \rangle \in \mathcal{T}$:*

1. *For every $l \in bindings(a)$ such that $\exists l_G \in g(a_G, l)$ where $l_G \notin s$, then $l \notin pre(a)$[3]*
2. *For every $l' \in bindings(a)$ such that $\exists l'_G \in g(a_G, l')$ and $l'_G \notin s'$ then $\nexists (c, e) \in eff(a)$ such that $\exists (c_G, e_G) \in g(a_G, c, e)$ where $(c_G \wedge s \nvdash \bot) \wedge (l'_G \in e_G)$*
3. *For every $l' \in bindings(a)$ if $\exists l'_G \in g(a_G, l')$ such that $l'_G \in s' \setminus s$ then $\exists (c, e) \in eff(a)$, where $\exists (c_G, e_G) \in g(a_G, c, e)$ such that $c_G \wedge s \nvdash \bot \wedge l'_G \in e_G$*
4. *For every $l' \in bindings(a)$ and $c \subseteq bindings(a)$ if $\exists (l'_g, c_G) \in g(a_G, c, l')$ such that $l'_G \in s' \setminus s$ and $c_G \wedge s \vdash \bot$ then $\nexists (c', e) \in eff(a)$ such that $c \subseteq c'$ and $l' \in e$*

Here too, the core Conditional-SAM algorithm remains the same, where *MustBeResult* and *PosAnte* may now contain parameter-bound literals that include UQVs. We assume that for each action the algorithm is aware of the types of objects that might include universal effects.

# 7 Experimental Results

We implemented Conditional-SAM and conducted experiments on six planning domains that include conditional effects. Specifically, we used the CityCar, Nurikabe, and Maintenance domains from the International Planning Competition (IPC) 2014 (Vallati et al. 2015); the Briefcase and Miconic are from AIPS-2000 (Bacchus 2001), and Satellite, which is an ADL version of the classical IPC (Long and Fox 2003) domain. [4]. Table 1 presents relevant information about the domains we experimented on. The column 'Domain' represents the domain's name that was experimented on, and the columns '$|A|$' and '$|P|$' present the number of actions and predicates in the domain respectively. The column '# U.E.' presents the number of universally conditional effects present in the domains (The satellite domain has conditional effects that do not contain UQVs) and the column '$n$' is the maximal number of antecedents for the conditional effects in the domains. The column $|\mathcal{T}|$ represents the size of the trajectories dataset of the domain. The column $|t|$ is the average number of action triplets in a trajectory (the standard deviation is displayed in brackets).

For each domain, we generated our problems dataset using a PDDL problem generator (Seipp, Torralba, and Hoffmann 2022).[5] Using the problem generator, we created a dataset of 100 problems that were used to create the trajectories. To solve the generated problems and create the input trajectories, we used two well-known classical planners that

---

[3]The first inductive rule enables learning universal preconditions

[4]All the domains are available in https://github.com/AI-Planning/classical-domains We provide an extensive explanation of the experimented domains as well as domains that were not used in our experiments in the supplementary material.

[5]Action costs were ignored in all domains we experimented on.

support ADL, Fast-Downward (FD) (Helmert 2006) using FF heuristic and context-enhanced additive heuristic, and Fast-Forward (FF) (Hoffmann 2001) with a Greedy BFS configuration. We restricted the solvers to solve the problems in up to 60 seconds. For the Nurikabe domain, only 52 problems were solved with our planners, resulting in a smaller dataset.

We split our dataset into train and test sets, trained Conditional-SAM on the trajectories in the train set, and used the generated action models to solve the test set problems. To validate the correctness of the generated plans we used VAL (Howey, Long, and Fox 2004). We followed a 5-fold cross-validation methodology by repeating each experiment 5 times, sampling different trajectories for learning and testing. All the presented results are averaged over the five folds. The experiments were run on a Linux machine with 8 cores and 16 GB of RAM.

| Domain | $|A|$ | $|P|$ | # U.E. | $n$ | $|\mathcal{T}|$ | $|t|$ |
|---|---|---|---|---|---|---|
| Satellite | 5 | 8 | 0 | 1 | 100 | 36.2 (6.1) |
| Maintenance | 1 | 3 | 1 | 1 | 100 | 5.6 (1.3) |
| Miconic | 3 | 6 | 2 | 2 | 100 | 46.7 (4.9) |
| Citycar | 7 | 10 | 1 | 1 | 100 | 19.7 (4.0) |
| Briefcase | 3 | 3 | 1 | 1 | 100 | 76.6 (15.6) |
| Nurikabe | 4 | 12 | 2 | 3 | 52 | 78.2 (5.17) |

Table 1: Statistics regarding the experimented domains.

| Domain | %S | %TO | %NS | %ERROR | Planner | $R_{sem}$ |
|---|---|---|---|---|---|---|
| Satellite | 100 | 0 | 0 | 0 | FD / FF | 0.99 |
| Maintenance | 100 | 0 | 0 | 0 | FF | 1.00 |
| Miconic | 100 | 0 | 0 | 0 | FF / FD | 1.00 |
| Citycar | 25 | 75 | 0 | 0 | FD | 0.99 |
| Briefcase | 16 | 25 | 0 | 59 | FF | 1.00 |
| Nurikabe | - | - | - | - | - | - |

Table 2: Experimental results for Conditional-SAM.

## Evaluation Metrics

We evaluated our algorithm using two metrics: the percentage of the test set problems solved using Conditional-SAM's learned model, and the correctness of the learned model using precision and recall measures. A test set problem is regarded as solved if one of the planners we used (FD and FF) was able to solve it with the learned action model. Measuring the *syntactic* precision and recall of the learned model, i.e., measuring the textual difference between the real and learned domains, may not represent the usefulness of the learned domain in solving problems. Instead, we measure the *semantic* precision and recall of the learned model's preconditions and effects, as follows. For each state in our trajectories, we try to apply the actions of the learned and real action models on the state. We measure the precision and recall based on which action is applicable in the tested states. Formally, the semantic precision and recall of the preconditions are:

$$P_{pre}^{sem}(a) = \frac{|app_{M^*}(a) \cap app_M(a)|}{|app_M(a)|}$$

$$R_{pre}^{sem}(a) = \frac{|app_{M^*}(a) \cap app_M(a)|}{|app_{M^*}(a)|}$$

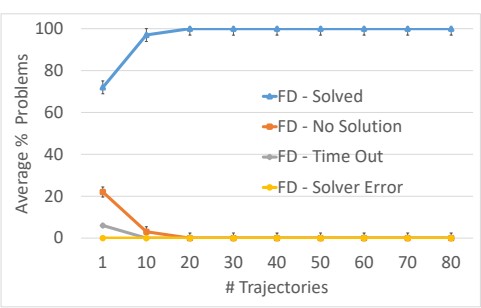

Figure 2: Satellite solving statistics.

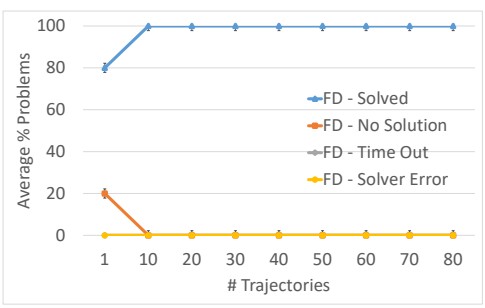

Figure 3: Miconic solving statistics.

Where $app_M(a)$ denotes the states in a set of trajectories where $a$ is applicable according to the action model $M$. Finally, average the results over all actions of the domain.

Since Conditional-SAM learns a safe action model, the semantic precision of the preconditions is always one. Furthermore, the Conditional-SAM's safety property indicates that whenever an action is applicable according to Conditional-SAM its effects are identical to the real domain's effects. Thus the precision and recall of the effects is always one as well. Thus, our evaluation below only presents semantic recall of the actions' preconditions.

## Results

Table 2 displays the experimental results with the maximal number of trajectories given as input. The column $\%S$ represents the percent of the problems that were solved by the planners, $\%TO$ represents the percentage of problems in

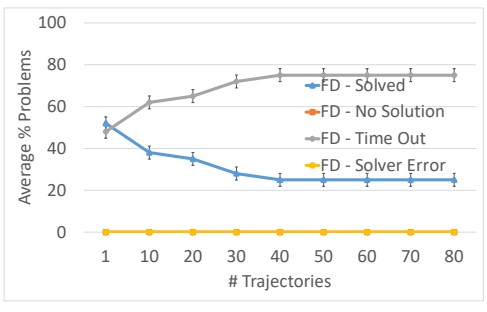

Figure 4: CityCar solving statistics.

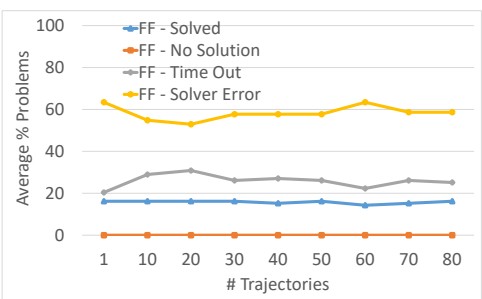

Figure 5: Briefcase solving statistics.

which the planner had timed out (i.e., reached our 60 seconds time limit), and $\%NS$ represents the percent of problems that were declared unsolvable with the learned domain. This is caused when the learned domain is too restrictive and thus some problems cannot be solved with it. The column $\%ERROR$ represents the percentage of problems in which the solver encountered an error while solving the test set problems. Such errors were caused when the planner was killed due to extensive resource consumption. The column *Planner* represents the planner that had the best performance and their results are being presented. Finally, the column $R_{pre}$ denotes the preconditions' semantic recall.

For the Satellite, Maintenance, and Miconic domains, all the test set problems were solved perfectly using the domain learned with Conditional-SAM. The results for CityCar and Briefcase are significantly worse, where the percent of problems that were solved are 25% and 19%, repsectively. A possible explanation for these results is that the learned domain returned by Conditional-SAM may contain complex universal preconditions. These preconditions affect the solvers in their ability to solve the test set problems. Indeed, every problem that was not solved in CityCar domain was not solved due to our timeout restrictions. Similarly, in Briefcase domain in many occasions, the planning process was terminated because it consumed too many resources (expressed in the $\%ERROR$ column). However, the calculated preconditions' semantic recall for both domains is 0.99. That suggests that while the learned domains may appear to be more complex than their original counterpart, they are nearly semantically identical. For the Nurikabe domain, Conditional-SAM could not solve any test problem with the learned domain. This may be because this domain has the largest antecedents ($n = 3$).

Figures 2, 3, 4, 5 presents the solving statistics as a function of the number of trajectories used to train Conditional-SAM for the domains Satellite, Miconic, CityCar, and Briefcase. We do not present the results for the Maintenance domain graphically since after a single trajectory all the test set problems were solved perfectly. We note that for both Satellite and Miconic domains, the percentage of problems increases monotonically, on the other hand, for CityCar domain, we observe a decrease in the number of solved problems as the number of trajectories increases. We note that with one trajectory the domain Conditional-SAM learns does not contain the action *destroy_road* which has universal effects. Without this action the domain was less complex and the planners

could solve more test set problems. Once the action was learned the domain became much more complex which resulted in the increase. Finally, we note that the poor results observed in the Briefcase domain were due to insufficient resources as can be observed in Figure 5 with 59% of the test set problems not being solved since the planning process was killed due to high resource consumption.

## 8    Supporting Disjunctive Antecedents

We focused on conditional effects where the result can appear only once in each action (Assumption 2. There are cases where such an assumption does not hold. For example, in our flu treatment action imagine that now the allergic reaction can appear if the patient has a rare blood type or if they are sleep deprived. In this case, our action would look as follows:

```
(:action treat-flu-symptoms-X
:parameters (?p - patient ?b_type - bloodType)
:precondition (and (has-flu ?p))
:effect
(and (when (is-rare-blood-type ?b_type)
          (allergic-reaction ?p))
     (when (sleep-deprived ?p)
          (allergic-reaction ?p))
     (and (not (has-flu ?p)))))
```

Figure 6: An action representing a flu medicine with a conditional effect.

In Figure 6 we present the action *treat-flu-symptoms-X* that now contains disjunctive antecedents. Supporting disjunctive antecedents requires altering the first assumption to address only the actions' preconditions and completely removing the second assumption. Removing the second assumption affects Conditional-SAM's fourth inductive rule since it no longer holds that if a conjunction of literals does not hold in a state, it cannot be and antecedent of the observed result. That is since now conditional effects might be disjunctive.

Supporting the new capability requires a minor change to the Conditional-SAM algorithm. We change the initialization process of *PosAnte* to have every possible CNF clause with up to $n$ antecedents, i.e., now *PosAnte* includes CNFs and not just conjunctions of literals. The rest of the algorithm is not affected. We note that this change highly increases the algorithm's complexity since now it has to eliminate every possible CNF expression before it can determine the correct set of antecedents. Note that the available benchmark domains do not contain disjunctive antecedents for conditional effects. Furthermore, due to its prohibitive complexity, we decided not to support disjunctive antecedents in Conditional-SAM and leave this functionality for future work.

## 9    Related Work

Several prior works learn action models from trajectories. The Action-Relation Modelling System (ARMS) (Yang, Wu, and Jiang 2007) algorithm learns a PDDL description of action models by extracting a set of weighted constraints from the input plan examples. The Simultaneous Learning and Filtering (SLAF) (Amir and Chang 2008) algorithm is a different algorithm for learning action models designed for partially observable deterministic domains. The Learning Object-

Centred Models (LOCM, LOCM2) (Cresswell, McCluskey, and West 2013; Cresswell and Gregory 2011) is another action model learning algorithm that analyzes plan sequences, where each action appears as an action name and arguments in the form of a vector of object names. FAMA (Aineto, Celorrio, and Onaindia 2019) is a state-of-the-art algorithm that learns action models with minimal state and action observability. FAMA can learn from gapped action sequences of actions, and in the extreme, FAMA can even learn when only given the initial and the final states as input.

The algorithms presented above learn action models that do not guarantee that the actions learned are applicable according to the agent's actual action model definition. Contrary to these algorithms, the SAM family of algorithms is designed to learn action models in a setting where execution failures must be avoided (Stern and Juba 2017; Juba, Le, and Stern 2021; Juba and Stern 2022). To this end, SAM generates a conservative action model. Planning with such an action model produces *sound* plans but may failu to find a plan even if such exists (i.e., it is *incomplete*).

To the best of our knowledge, there is no work focusing on learning safe action models with conditional effects. In (Oates and Cohen 1996), the authors created an algorithm that can learn planning operators for STRIPS (Fikes and Nilsson 1971) by interacting with the environments and performing random actions, and using search techniques to learn the context-dependent operators. This approach uses random walks which are costly in case that the agent cannot recover from failures. Furthermore, the resulting action model generated is grounded while our approach learns a lifted PDDL domain. In (Zhuo et al. 2010), the authors' main focus was learning action models with quantifiers and implications. The authors also proved that their algorithm could learn simple conditional effects with only one antecedent. Since the authors' goal is to *reduce* the domain compilation time for domain experts, the domains their algorithm outputs may be incomplete or even wrong. This means that their algorithm does not work in mission-critical settings.

## 10    Conclusions and Future Work

In this work, we presented Conditional-SAM, an algorithm that can learn action models for domains that include conditional and universal effects. We showed that Conditional-SAM learns a safe action model w.r.t the real unknown action model and runs in reasonable time. Moreover, we presented tight sample complexity results, showing that Conditional-SAM is, in a sense, asympotitcally optimal. Our experimental results show that using a small number of trajectories, Conditional-SAM learns an action model that solves the test set problems. In future works, we aim to explore methods to improve the algorithm's scalability and support domains with more expressive conditional effects that might contain unbounded disjunctive antecedents or even include numeric conditions and effects.

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
