# OpenReview forum: "Safe Learning of PDDL Domains with Conditional Effects"
_icaps-conference.org/ICAPS/2024/Conference — ICAPS 2024_

### Official Review · Reviewer_RZKw · 2024-01-12

**Significance And Importance:** 2
**Soundness:** 3
**Novelty:** 2
**Clarity:** 3
**Overall Evaluation:** 1
**Confidence:** 2

**Weaknesses:**

1: Minor weaknesses that are easily fixable.

**Contributions Of The Paper:**

The extension of the SAM learning algorithm is presented in this paper.  The paper wants to do safe learning of PDDL domains.  The novelty is to allow conditional effects.  The authors propose an algorithm for this, give some theoretical results about complexity, and test it empirically.

It was not easy for this reviewer to follow the technical development of the algorithm.

The paper seems to be quite complete, and at a surface level sound.  The novelty seems to be adequate for a conference paper.


Minor comments:

on page 1, there is an extra space after the closing paren in the abstract

there is a double closing paren before ".  Defining"

at the start of section 7, "domain. 4." should be "domain.4"

at the start of section 8, there is a missing closing paren

in the last para of section 9, there are some \cite's which should be \citet's

at the start of section 10, "w.r.t" misses a final full stop

**Ethical Considerations:**

(1) Not Applicable: The paper does not have any ethical considerations to address

**Nomination For Best Paper:**

No

**Questions For Authors:**

i do not have any questions.

**Reproducibility:**

3: Authors describe the implementation and domains in sufficient detail.

**Strengths Of The Paper:**

See above.

**Weaknesses Of The Paper:**

See above.

---

> ### Author Rebuttal · Authors · 2024-01-28
>
> We thank the reviewer for their comments and intend to fix all the issues raised in the comments.

---

### Official Review · Reviewer_6JiC · 2024-01-19

**Significance And Importance:** 2
**Soundness:** 3
**Novelty:** 3
**Clarity:** 3
**Overall Evaluation:** 2
**Confidence:** 3

**Weaknesses:**

1: Minor weaknesses that are easily fixable.

**Contributions Of The Paper:**

This paper describes the safe learning of classical planning domains with conditional effects.  The approach builds on similar approaches developed for safe learning of probabilistic planning domains, and older approaches for domain model learning for classical domains from plan traces..  The key innovations are an approach for learning ADL-style domains with conditional effects from traces, provided a bound on the number of terms in the antecedent of the conditional effect is provided.  A bound on the number of samples needed to ensure safety is provided.  An interesting aspect of the approach is that the learned model need not have exactly the same form as the true (and unknown) model, but nevertheless ensures that a conditional effect is only added when appropriate. A theoretical analysis (largely left to supplemental material) shows time, space and sample complexity to ensure safety (essentially the probability an effect isn’t added is shown to be very small if enough samples are provided under proper independence assumptions.). The approach is generalized to a variety of related problems like lifted action models and universally quantified effects.

**Ethical Considerations:**

(1) Not Applicable: The paper does not have any ethical considerations to address

**Nomination For Best Paper:**

No

**Questions For Authors:**

are the theorems and table 1 really only counting predicate symbols and not true ground predicates?  Yes, they are, because classical domains only propagate bindings, and don’t have complex constraints on the parameters.  Presumably if we move to a more sophisticated language like SAS or a Timeline based framework, the resulting theorems need to be generalized to capture the number of ground fluents?

**Reproducibility:**

4: Authors promise to release code and domains (whichever apply).

**Strengths Of The Paper:**

The paper solves an interesting problem, that of learning domains from plan traces.  The paper buiids on prior approaches to learning and applies them to planning with conditional effects.  The major algorithms are described; theoretical results are provided but details deferred to supplemental material.  The empirical analysis is well done. The paper is somewhat dense but overall well organized and well written.

**Weaknesses Of The Paper:**

The paper does not include empirical analysis of model build time.  It would be nice to see these; does model building take hours?  Days?

The paper also has a few minor notational inconsistencies (see detailed comments to follow.)
the citation for PDDL should be McDermott, not Ghallab
p. 3: when you say you add condition l V  NotAnte V Ante when there is ambiguity, perhaps explain a little more why you add NotAnte to the disjunction.  Does this protect against the possibility that l is added unconditionally due to ambiguity in the set PostAnte?
p. 2,4: the notation in Thm 4.3 and 4.4. uses the symbol F, pls add |L| = 2|F| somewhere so reader won’t be confused switching from lemmas to theorems.
p. 6: the tables use P for number of predicates instead of L (or F).  which is it?  Looks like F is what you want here.
p. 6: how does the number of trajectories (100) compare to the theoretical sample trajectory complexity?  If I understand the theorems and table 1 right, let’s pick Satellite. A=5, F=8, so the term ’40’ appears twice in thm 4.3 with coefficients > 1 (never mind that nasty F^n term!) and with the leading term 1/\epsilon that we want to be small.  So I’d expect a vast number of trajectories.  I am somewhat surprised at the high quality of results with so few trajectories.  comments?

---

> ### Author Rebuttal · Authors · 2024-01-28
>
> We thank the reviewer for the helpful comments.
>
> Re. the reviewer’s question, indeed we learn a lifted domain and currently do not support more complex conditions between parameters than explained in the paper. We are not sure why the reviewer is suggesting that moving to SAS+ would require learning at the grounded level.
>
> Re. the reviewer's other comments:
> 1. We intend to fix all the minor issues (typos, etc.) raised by the reviewer. Thank you for pointing them!
> 2. The actual time it takes to build the action model varies in the number of input trajectories since the algorithm iterates over every action triplet. Empirically, for most domains, the process is quick, i.e., a matter of seconds.
> 3. Re. the additional NotAnte:  Safety requires knowing whether or not l appears in the post-state. If l was not in pre, then we need to know whether or not l is an effect. NotAnte allows us to execute the action in cases where we know l can't be an effect.
> 5. The reviewer is correct in stating that P  in the table should in fact be `F`. We will fix this as suggested.
> 6. Note that 100 trajectories contain more than 100 action triplets. Furthermore, in the Satellite domain, the number of antecedents for each conditional effect is at most one, which makes the learning task easier.

---

### Official Review · Reviewer_rTna · 2024-01-22

**Significance And Importance:** 2
**Soundness:** 3
**Novelty:** 3
**Clarity:** 3
**Confidence:** 4

**Weaknesses:**

0: Minor weaknesses requiring some work to be addressed for the paper to be accepted.

**Contributions Of The Paper:**

This work presents an improved (borderline incremental) version of SAM-learning that incorporates learning conditional and universally quantified preconditions and effects. The paper presents a thorough theoretical analysis of the complexity and the number of samples needed to learn the correct action models with different properties.

**Ethical Considerations:**

(1) Not Applicable: The paper does not have any ethical considerations to address

**Nomination For Best Paper:**

No

**Overall Evaluation:**

-1: (weak reject)

**Questions For Authors:**

1. In table 1, |A| and |P| corresponds to the number of lifted actions and predicates or grounded actions and predicates?
2. A different planner (FF) for Briefcase is presented compared to FD for other domains in Figure 5. Lines 581-583 do mention the presented results are for best-performing planners, but how much worse is the performance on FD? If the difference is substantial, what might be the reason for it?
3. About assumption 2 mentioned in lines 189-190, can an action have the following conditional effects?
$(p1 \land p2) \rightarrow (p3 \land p4 \land p5)$
$(p2 \land p6) \rightarrow (p4 \land p8)$
4. Please comment on the algorithm’s scalability in terms of number of antecedents (n).
5. Were there any experiments done to test the correct learning of UQVs? If yes, was it separated for quantifies preconditions and effects?

**Reproducibility:**

3: Authors describe the implementation and domains in sufficient detail.

**Strengths Of The Paper:**

1. The theoretical treatment of the setup is impressive and one of the significant strengths of the paper.
2. The algorithm is easy to follow, with a good explanation for the grounded version.

**Weaknesses Of The Paper:**

1. Lines 119-121 describe preconditions as conjunction of literals whereas lines 291-298 move to a disjunctive precondition. Not sure if this is a typo, but this is a major discrepancy that should be fixed.

2. I am a bit confused about the assumption 2 mentioned in lines 189-190. Does this mean that for an action we cannot have these conditional effects? Here p4 appears in both the effects.
$(p1 \land p2) \rightarrow (p3 \land p4 \land p5)$
$(p2 \land p6) \rightarrow (p4 \land p8)$

3. The clarity of the paper would increase a lot if the line numbers corresponding to the algorithm were mentioned in the section “Conditional-SAM Algorithm.” For example, It took me some time to understand that lines 10-13 correspond to rule 3, not just lines 10-11.

4. A major weakness in clarity is about how the algorithm will be updated corresponding to Sections 5 and 6. Lifted operator learning is not trivial, and lines 435-437 and 492-496 handwaves around this complex process. I strongly suggest adding the algorithm with modifications, at least in the supplementary, with these changes. Right now, I have to guess what the correct changes would be, and it would be great not to have to guess them.

5. The high number of antecedents (n=3) results in fewer solutions, as mentioned in lines 602-604. And n=3 does not seem that high a number given that there are 12 predicates (or propositions?) in the domain. This points to a problem in the scalability of the approach. And since there is only one domain with n=3, it is difficult to evaluate the algorithm’s scalability.

6. Experiments on UQVs might be missing. Looking at the supplemental, only airport-adl has "Existential preconditions/effects". And the 4 IPC domains whose results are presented do not have this feature. So it is possible that this feature of the algorithm was not tested at all in the empirical evaluation.

7. Presentation Suggestion: I strongly recommend combining Figures 2-5 into a single figure and using a common x or y-axis. It would greatly improve the presentation of the results.

---

> ### Author Rebuttal · Authors · 2024-01-28
>
> Re. the weaknesses pointed by this reviewer:
> 1.	The real model does not have actions with disjunctive antecedents (assumption 2) but  the action model we learn may include such actions. See line 315. This highlights the fact that the learned model can be different from the real action model that we are trying to learn.
> 2.	(p1∧p2)→ (p3∧p4∧p5) and (p2∧p6)→(p4∧p8) can be translated to: (p2∧p6)→p4 or (p1∧p2)→p4. Thus, these are disjunctive conditional effects, which we do not support.
> 3.	We will add line numbers as suggested.
> 4.	Most of the algorithm remains the same as in the grounded case with the exception of the inductive learning rules mentioned in the text. We intend to add a pointer to such a pseudo-code as suggested.
> 5.	Indeed, the runtime grows exponentially with the number of UQVs, n, since the possible antecedents for each possible result are all the possible conjunctions of size up to n of predicates with UQVs. Unfortunately, the current standard benchmarks do not have domains with a larger number of UQVs.
> 6.	We added a list of domains to the supplementary material to name the ADL domains available with their properties. In addition, the table is intended to give the reader an indication of the domains that were not used in our experimentations and the reason for doing so. We intend to make it clearer in the final version.
> 7.	We intend to fix the images according to the reviewer’s suggestion.
>
> Answers:
> 1.	|A| and |P| are the number of lifted actions and predicates.
> 2.	In these domains FD could not solve any of the problems. Studying this phenomenon is worthwhile, but our focus is on the behavior of planners with different (learned) action models rather than the behavior of different numeric planners with the same action model.
> 3.	See comment (2) above.
> 4.	See comment (5) above. Briefly, the runtime is exponential in n.
> 5.	There are experiments for universal effects using universally quantified variables, but there are no experiments for universal preconditions that do not contain additional features that we currently do not support.

---

### Meta-Review · Area_Chair_YDKe · 2024-02-06

**Recommendation:** Accept (Poster)
**Confidence:** 3

**Metareview:**

The paper extends the SAM algorithm for learning PDDL models from observations to conditional effects and to lifted representations (including universally quantified effects). In the discussion, all authors still agreed that they support or at least are ok with acceptance.

Pros:
- overcoming earlier limitations of SAM
- theoretical analysis of required number of samples
- empirical evaluation overall well-done (mixed results)

Cons:
- somewhat incremental
- clarity should be improved (specific hints in reviews)
- limited scalability (already an issue of original SAM)
- details of theoretical results only in supplemental material
- universal quantification not sufficiently represented in benchmarks

**Ethical Considerations:**

(1) Not Applicable: The paper does not have any ethical considerations to address